# Combining Different Docking Engines and Consensus Strategies to Design and Validate Optimized Virtual Screening Protocols for the SARS-CoV-2 3CL Protease

**DOI:** 10.3390/molecules26040797

**Published:** 2021-02-04

**Authors:** Candida Manelfi, Jonas Gossen, Silvia Gervasoni, Carmine Talarico, Simone Albani, Benjamin Joseph Philipp, Francesco Musiani, Giulio Vistoli, Giulia Rossetti, Andrea Rosario Beccari, Alessandro Pedretti

**Affiliations:** 1Dompé Farmaceutici SpA, Via Campo di Pile, 67100 L’Aquila, Italy; candida.manelfi@dompe.com (C.M.); carmine.talarico@dompe.com (C.T.); andrea.beccari@dompe.com (A.R.B.); 2Computational Biomedicine, Institute for Neuroscience and Medicine (INM-9) and Institute for Advanced Simulations (IAS-5), Forschungszentrum Jülich, 52425 Jülich, Germany; j.gossen@fz-juelich.de (J.G.); s.albani@fz-juelich.de (S.A.); benjamin.joseph@rwth-aachen.de (B.J.P.); g.rossetti@fz-juelich.de (G.R.); 3Faculty of Mathematics, Computer Science and Natural Sciences, RWTH Aachen, 52062 Aachen, Germany; 4Dipartimento di Scienze Farmaceutiche, Università degli Studi di Milano, Via Mangiagalli, 25, I-20133 Milano, Italy; silvia.gervasoni@unimi.it (S.G.); giulio.vistoli@unimi.it (G.V.); 5Laboratory of Bioinorganic Chemistry, Department of Pharmacy and Biotechnology, University of Bologna, 40127 Bologna, Italy; francesco.musiani@unibo.it; 6Jülich Supercomputing Center (JSC), Forschungszentrum Jülich, 52425 Jülich, Germany; 7Department of Hematology, Oncology, Hemostaseology and Stem Cell Transplantation University Hospital Aachen, RWTH Aachen University, Pauwelsstraße 30, 52074 Aachen, Germany

**Keywords:** SARS-CoV-2, 3CL-Pro, antivirals, virtual screening, docking simulations, drug repurposing, consensus models, binding space, isomeric space

## Abstract

The 3CL-Protease appears to be a very promising medicinal target to develop anti-SARS-CoV-2 agents. The availability of resolved structures allows structure-based computational approaches to be carried out even though the lack of known inhibitors prevents a proper validation of the performed simulations. The innovative idea of the study is to exploit known inhibitors of SARS-CoV 3CL-Pro as a training set to perform and validate multiple virtual screening campaigns. Docking simulations using four different programs (Fred, Glide, LiGen, and PLANTS) were performed investigating the role of both multiple binding modes (by binding space) and multiple isomers/states (by developing the corresponding isomeric space). The computed docking scores were used to develop consensus models, which allow an in-depth comparison of the resulting performances. On average, the reached performances revealed the different sensitivity to isomeric differences and multiple binding modes between the four docking engines. In detail, Glide and LiGen are the tools that best benefit from isomeric and binding space, respectively, while Fred is the most insensitive program. The obtained results emphasize the fruitful role of combining various docking tools to optimize the predictive performances. Taken together, the performed simulations allowed the rational development of highly performing virtual screening workflows, which could be further optimized by considering different 3CL-Pro structures and, more importantly, by including true SARS-CoV-2 3CL-Pro inhibitors (as learning set) when available.

## 1. Introduction

Coronaviruses (CoVs, subfamily Coronavirinae, and family Coronaviridae) are enveloped viruses consisting of a single positive-strand RNA that can infect humans where they may cause respiratory, gastro-intestinal, and neurological disorders. A recently identified new coronavirus appeared in Wuhan, China [1], at the end of 2019 to cause a world-wide pandemic crisis in the present times [2]. This is mainly responsible for a pneumonia-like illness that shows severe threats and often requires patient hospitalization with a global lethality equal to 2.1% (data updated to 16 January 2021). By considering the caused illness and its similarity with the SARS coronavirus (SARS-CoV, genome equal to 82%), the virus was called SARS-CoV-2 (severe acute respiratory syndrome-coronavirus-2), while the induced disease was termed Covid-19 [3].

The SARS-CoV-2 genome [4] encodes for structural proteins that are required for viral entry (such as the Spike glycoprotein); non-structural proteins, which comprise enzymes endowed with protease, methyltransferase, helicase, and polymerase activities; and accessory proteins [5], the role of which should be yet fully clarified [6,7]. The non-structural proteins as well as the proteins required for the viral interaction with the host cells represent potential drug targets [8]. In detail, the SARS-CoV-2 replicase gene encodes for two overlapping polyprotein structures (i.e., ppa1a and ppa1ab), which are cleaved in the functional proteins required for viral replication and transcription by a 33.8 kDa protease called M^Pro^ or 3-chymotrypsin like protein, 3CL-Pro [9]. Interestingly, 3CL-Pro itself is comprised within the ppa1a and ppa1ab polyproteins, and indeed, the first enzymatic step involves the autolytic cleavage to liberate 3CL-Pro. This protease is active as a homodimer and shows a peculiar specificity, which differs from the close proteases of the host cells, since it shows a unique substrate preference for glutamine at the P1 site [10].

Such a substrate specificity combined with its key role in the viral life cycle renders such a protein an attractive target to develop anti-SARS-CoV-2 drugs. Thus, a relevant number of publications focused on the identification of potential SARS-CoV-2 3CL-Pro inhibitors appeared in the literature during the last few months [11]. Several studies comprised structure-based virtual screening (VS) campaigns especially targeted for drug repurposing. For example, Gimeno et al. [12] reported a consensus repurposing study which combines three docking programs (Glide, FRED, and AutoDock/Vina). Only the ligands showing favorable and equivalent binding modes in all three programs were considered as actives. Elmezayen et al. combined docking calculations with molecular dynamics (MD) simulations to better investigate the stability of the retrieved hits by evaluating the corresponding free energy using the MM-PBSA method [13]. Again, Meyer-Almes described a method in which molecular docking, ∆G energy calculation, and analysis of the protein ligand interaction fingerprints (PLIF) are combined to increase the predictive performances [14]. Other studies involved computational protocols based on a single docking tool and extended the virtual screenings also to drug-like molecules included within the ZINC database [15] as well as to databases of natural compounds [16]. However, and due to the lack of known SARS-CoV-2 3CL-Pro inhibitors, all the reported VS strategies cannot be optimized and validated a priori, and thus the reliability of the retrieved hits has to be corroborated by combining different computational approaches.

The novelty of the present study is that the known SARS-CoV 3CL-Pro inhibitors can be added to the screened databases and used as a training set of active molecules to evaluate the performances of the applied computational strategies. Although the efficacy of the here simulated SARS-CoV 3CL-Pro inhibitors was not experimentally confirmed against SARS-CoV-2 3CL-Pro, such a choice is justified by the very high conservation degree between the two proteases (SARS-CoV and SARS-CoV-2 3CL-Pro differ only for 12 residues out of 306), which is suggestive of a very similar binding mode within the two enzymatic pockets, as also confirmed by a recent computational analysis [17].

The employment of the known SARS-CoV 3CL-Pro inhibitors as active molecules in the here reported simulations is also confirmed by the recently resolved structures of SARS-CoV-2 3CL-Pro enzyme in complex with known SARS-CoV 3CL-Pro inhibitors, which show ligand arrangements superimposable to those already seen within SARS-CoV 3CL-Pro. A compelling example is offered by the SARS-CoV-2 3CL-Pro structure bound to the potent TG-0205221 inhibitor (PDB Id: 7C8T), which is completely comparable with the corresponding SARS-CoV 3CL-Pro complex (PDB Id: 2GX4, see below). On one hand, such a resolved structure indirectly confirms the potential activity on SARS-CoV-2 3CL-Pro of the most active known inhibitors of SARS-CoV 3CL-Pro. On the other hand, this structure emphasizes the possibility to employ known SARS-CoV 3CL-Pro inhibitors to guide the design and/or selection of promising hits for SARS-CoV-2 3CL-Pro enzyme.

Hence, the study involves extended docking simulations based on a purposely collected database in which a set of 478 molecules active against SARS-CoV 3CL-Pro with a pIC50 values greater than 5.0 were dispersed within a database of about 13,000 safe in man molecules. In this way, the performed simulations can be exploited to design, optimize, and validate targeted VS protocols by comparing and combining different docking tools (i.e., Glide, LiGen, Plants, and Fred) and different post-docking analyses. Moreover, the best performing approaches can also be used to identify potential hits against SARS-CoV-2 3CL-Pro.

Furthermore, the analysis of the performed VS campaigns involves the combination of the already proposed binding space with the here developed isomeric space. The concept of binding space was recently proposed by some of us to account for the multiple binding modes a ligand can assume within the binding pocket by simultaneously considering more than one pose for each ligand [18]. Similar to property space [19], the binding space can be defined by parameters that encode for both the average and the spread of the scores values, the latter being described by ranges or standard deviations. Previous studies involving both correlative analyses and VS campaigns revealed that the binding space concept can elicit significant improvements in the predictive power compared to the best score values [20].

In this study, the same rationale inspiring the definition of the binding space was applied to define the corresponding isomeric space. In detail, the ligands existing in different tautomers, stereoisomers, or protonation forms are simulated by docking all possible states, and the resulting docking scores are utilized for calculating the corresponding space parameters (i.e., score averages and ranges). As detailed under Methods, the study investigates both the binding and isomeric spaces by considering four possible combinations: (1) without including the parameters of both the binding and isomeric spaces but considering only the best scores for each compound (namely, the canonical conditions); (2) by including the parameters of the sole binding space to confirm the encouraging results already pursued in previous studies; (3) by including the parameters of the sole isomeric space to assess its specific relevance; and (4) by variously combining the parameters of both spaces to investigate their synergistic role. All so-computed score parameters were then utilized to develop consensus models by using the enrichment factor optimization (EFO) approach [21], which generates linear combinations of the considered descriptors and proved particularly efficient in previous VS benchmarking studies [22,23].

## 2. Results

### 2.1. Simulated Dataset

Docking simulations involved a database of 13,535 unique molecules including 478 known SARS-CoV inhibitors with pIC50 > 5, taken from literature, which were dispersed within a dataset of 13,057 safe in man compounds. Among the 478 known SARS-CoV inhibitors, 193 compounds are endowed with a pIC50 > 6. These two inhibition thresholds (i.e., pIC50 > 5 and pIC50 > 6) allow the definition of two different (despite partly overlapped) sets of active molecules, and the generation of all analyzed consensus models was repeated by considering the two resulting training sets of SARS-CoV inhibitors. Furthermore, 4515 molecules (i.e., a third of the complete database) required the generation of 14,013 multiple states/isomers with an average of 3.1 isomers per molecule. In this way, the screened database overall included 23,054 ligands.

Due to the different docking protocols and the criteria by which the generated poses were considered as acceptable by each docking tool (as discussed under Methods), the number of docked molecules varies among the docking simulations as summarized in Table 1. In detail, Table 1 shows that the number of simulated molecules by the four tested programs decreases with the following trend: PLANTS > LiGen > Fred > Glide. The number of docked active molecules parallels the same trend except for Glide, which considers a higher number of active molecules while simulating a lower number of ligands (compared to Fred). The same trend is exhibited by the number of molecules simulated by considering multiple states by the four docking engines.

When considering the number of monitored poses, it should be noted that PLANTS, Fred, and Glide generated 10 poses per ligand, while LiGen provided 10 poses per unique molecule. This implies that for molecules existing in multiple states, LiGen selected the 10 best poses regardless of the involved isomers. Hence, the number of analyzed poses for LiGen is significantly lower than that of the other docking engines.

Despite the described differences, a common dataset composed of those molecules that are simulated by all the utilized docking tools can be extracted. This comprises 10,110 ligands (about 75% of the full database) and includes 353 active compounds (i.e., with pIC50 > 5). Such a common database will be used to develop mixed consensus models by combining the docking results from different tools and to compare the best ranked molecules by the four docking tools.

### 2.2. Results by PLANTS

Table 2 compiles the best enrichment factors in the top 1% (EF 1%) based on the PLANTS simulations as well as the resulting enhancements (as percentage values) induced by the inclusion of the explored space descriptors and by the generation of linear consensus equations. In all reported analyses (Table 2, Table 3, Table 4 and Table 5), the consensus models are developed by linearly combining from two to five docking scores using the EFO algorithm (as detailed under Methods). As mentioned above, the analyses are repeated by considering the two sets of active molecules as collected considering two different inhibition thresholds (i.e., pIC50 > 5 and pIC50 > 6). 

A preliminary consideration, which affects all these analyses, concerns the unavoidably biased comparison between binding and isomeric spaces. Indeed, the former involves all the screened molecules for which different poses were generated, while the latter affects only those molecules existing in multiple states, which represent about 1/3 of the entire dataset (see above). Specifically, the PLANTS simulations involved 31 compounds existing in multiple states out of 193 for the actives with pIC50 > 6 and 136 out of 478 for the molecules with pIC50 > 5.

Thus, one may understand the different effects exerted by the inclusion of the isomeric space parameters for the two considered pIC50 thresholds. Indeed, the number of examined isomers with pIC50 > 5 is high enough and the isomeric space affords an appreciable enhancing effect, which is greater than that exerted by the binding space. In contrast, the limited number of compounds existing in multiple states with pIC50 > 6 reduces the enhancing effect exerted by the isomeric space. The inclusion of the binding space parameters induces significant improvements, especially with pIC50 > 6, where it is more effective than the isomeric space, especially when considering the best obtained models.

Regardless of the adopted strategy, the combination of the two monitored spaces appears to be productive only with pIC50 > 5, where it affords average improvements greater than those reached by the individual spaces. In contrast, the space combination exerts very modest effects with pIC50 > 6, since the resulting best models do not exceed the performances reached by the sole binding space. Overall, PLANTS affords encouraging performances that benefit from both the inclusion of the space descriptors and the generation of consensus linear equations. The latter induces an average improvement of around 39% with the best improvements around 70%.

While avoiding the systematic analysis of the generated consensus equations, attention will be here focused on the occurrence of the various scores in all generated models. The interested reader can find all computed consensus equations with the resulting performances for all docking programs and all performed analyses at http://www.exscalate4cov.network/. Appendix A reports the occurrence of the diverse scores in the 20 best consensus models generated by the EFO algorithm in all the 10 analyses included in Table 2 (for a total of 200 equations). In detail, Appendix A reveals the remarkable role played by the XScore [24] and PLANTS [25] scoring functions (the latter here comprising also the primary scores). Appendix A also evidences the interesting role played by the scores encoding for non-polar interactions as exemplified by molecular lipophilic potential (MLP) [26] and the VEGA-based scores [27], which here correspond to the Lennard–Jones interaction energies as computed by using the CHARMM and CVFF force fields. Concerning the type of score values (in the models including the space descriptors), the best values play a prevailing role, while spread and mean values have a lesser and rather similar incidence.

### 2.3. Results by LiGen

As described under Methods, the analyses based on the LiGen simulations involved the primary scores, the scoring functions derived by rescoring analyses, and 30 pharmacophoric distances (PH) as generated by the LiGen software. Table 3 reports the performances reached by the LiGen results as well as the performance enhancements due to both the inclusion of the space parameters and the generation of consensus models (as percentage values).

The first key observation concerns the very limited role played by the isomeric space for both pIC50 thresholds. Such a result cannot be ascribed to a too low number of active molecules existing in multiple states, since they roughly correspond to those already considered by PLANTS (i.e., 132 out of 472 with pIC50 > 5 and 29 out of 190 with pIC50 > 6). Instead, this finding can be explained by considering that limiting the analysis to 10 poses per unique molecule does not allow an extensive exploration of the isomeric space. Moreover, the scoring functions and the search algorithm implemented by LiGen might be not so sensitive to the isomeric differences among the simulated ligands. When considering the overall satisfactory performances reached by LiGen (see below), the poor results of the isomeric space can be positively evaluated, since they suggest that this docking program can conveniently perform VS campaigns without requiring excessive isomeric expansions of the screened databases with a beneficial reduction of computational time and complexity.

In contrast, the inclusion of the binding space descriptors markedly improves the EF1% values with an enhancement effect, which is particularly evident with pIC50 > 6, where the binding space leads to almost doubled EF1% values. When considering that the LiGen program produced the lowest number of poses, the remarkable results yielded by the binding space parameters emphasize that this reduction is yet able to extract the significant poses, thus minimizing redundant results and reducing the computational costs. Based on the different effects induced by the two analyzed spaces, one may understand why their combinations are unproductive since they yield EF1% values, which are comparable with those obtained by the sole binding space.

The LiGen results seem to particularly benefit from the linear combination of diverse scoring functions: this effect is noticeable in all the experiments, as confirmed by the average increases around 30%, but is particularly remarkable with pIC50 > 6, where the consensus models lead to the doubling of the corresponding EF1% value. Taken together, LiGen provides satisfactory performances, reaching EF1% values around 25 for pIC50 > 6 and with the inclusion of the binding space descriptors.

Since the LiGen analyses also comprise the pharmacophoric distances, a specific experiment was performed to investigate their specific role. In detail, the analysis involved the generation of two additional five-variable consensus equations for each pIC50 threshold. The first model was based on the pharmacophoric distances only and the second on all computed scoring functions except for the pharmacophoric distances. The resulting EF1% values (see Table 3) highlight an overall synergistic effect between pharmacophoric distances and scoring functions. Nevertheless, Table 3 reveals different behaviors depending on the considered inhibition threshold. The pharmacophoric distances play indeed a prevailing role with pIC50 > 5, while the score values assume a more marked role with pIC50 > 6.

Appendix A compiles the incidence of the various docking scores in all the generated consensus equations and reveals the prevailing role played by the pharmacophoric distances as computed by LiGen. Apart from these distances, only PLANTS scores show a significant relevance, while the scores encoding for specific interactions exhibit negligible roles. As seen above, the best values confirm their prevailing role, but here the spread values reveal an incidence that is almost double compared to the score means.

### 2.4. Results by Fred

Table 4 compiles the EF1% values and the corresponding enhancements as derived by the Fred docking results. The first key consideration is that the single scoring functions provide satisfactory results even without space parameters and consensus combinations as seen with pIC50 > 6. The notable performances elicited by simple docking scores can explain why the inclusion of the descriptors for both explored spaces exerts only limited effects. Similar to what was seen for LiGen, the isomeric space also plays here an almost negligible role, while the binding space parameters yield modest improvements by increasing the corresponding EF1% values of about 10% in the best consensus models. This result can be explained by considering that the beneficial effects of such post-docking procedures (such as rescoring, the inclusion of binding and isomeric spaces, and consensus approaches) depend on the room for improvement that the simple docking simulations show, and they inevitably decrease when the standard docking results already reveal notable predictive powers.

The docking simulations by Fred involved several active molecules existing in multiple states roughly comparable to those simulated by Plants and LiGen (despite the lowest number of total simulated isomers: i.e., 118 out of 411 with pIC50 > 5 and 27 out of 138 with pIC50 > 6). Hence, the modest relevance of the isomeric space suggests that this docking program does not require significant isomeric expansion of the screened databases to perform reliable VS campaigns. Similarly, the modest effect also exerted by the binding space suggests that the poses generated by Fred show a limited variability and tend to be focused around the best (and reasonably reliable) pose. In this way, the score averages roughly correspond to the best values, and the score spreads lose most of their relevance. As discussed above for the LiGen results, these insensitivities can be seen as positive features, which allow docking simulations to be performed without significant isomeric expansions of the screened datasets and reducing the number of poses computed per ligand thus minimizing the computational costs. Despite the modest effects exerted by the single spaces alone, their combinations yield encouraging results, especially with pIC50 > 5.

Appendix A shows the relative incidence of the computed scoring functions, as seen in all the consensus models generated by Fred. The key observation is that here PLANTS and XScore functions show an almost exclusive role, which minimizes the relevance of all other docking scores. While considering the unsatisfactory results afforded by space parameters and consensus strategies (see above), these results indicate that at least the rescoring calculations played a key role in enhancing the predictive power of the Fred simulations. About the type of score values, the best scores represent the most abundant values, while the mean values play a more relevant role here than the score ranges.

### 2.5. Results by Glide

Table 5 compiles the EF1% values (and the corresponding enhancements) as derived by using Glide. Despite the lowest number of active compounds simulated in multiple states (i.e., 105 out of 415 with pIC50 > 5 and 24 out of 164 with pIC50 > 6), Table 5 shows the markedly beneficial effect exerted by the isomeric space with both pIC50 thresholds. Although the binding space also elicits encouraging EF1% improvements for both thresholds, the isomeric space affords better results in terms of both reached EF1% values and relative performance enhancements. As already evidenced by previous studies [28], these results suggest that the search algorithms and the scoring functions implemented by Glide are strongly dependent on isomeric differences and invite the exhaustive expansion of the screened databases by considering as many as reasonable isomers/states, even when the number of simulated molecules existing in multiple states is relatively low.

While in the previous analyses the space combinations provided limited enhancements without significant differences between the two applied strategies; here, the joint combination of the two spaces yields remarkable performance enhancements compared to the single spaces with both pIC50 thresholds. In contrast, the merging combination approach appears to be constantly unproductive. These results suggest that the Glide-based docking scores are able to properly account for both multiple states (by isomeric space) and multiple binding modes (by binding space). Such a sensitivity implies that the two explored spaces encode for different information, and thus their descriptors can be synergistically combined, but cannot be fused into a unique space, the descriptors of which would detrimentally confuse the specific roles of the two spaces (as seen with the merging combination). Taken globally, the Glide simulations provide satisfactory predictive performances, which are, on average, comparable with those offered by LiGen with best EF1% values ranging between 25 and 30. As discussed above, the obtained performances markedly benefit from both explored spaces (and their joint combination) as well as from the development of consensus linear equations as assessed by an overall improvement of about 20% (with best results around 25%).

Appendix A reports the incidence of the various scoring functions in all the equations developed by using the Glide results and emphasizes the relevant role played by the 23 different primary Glide scores [29]. Notably, they appear to be particularly abundant in those experiments, which afforded the best performances. The equations seem to benefit from the inclusion of scoring functions encoding for non-polar interactions. Indeed, the VEGA-based scores, which here correspond to the CHARMM- and CVFF-based Lennard–Jones interaction energies, and the MLP values show an overall incidence of about 20%. Concerning the score types, the best values represent about 50%, and mean and spread values are equally abundant.

### 2.6. Overall Comparison

Although the differences in the protocols adopted and in the successfully docked molecules by the four tested docking programs prevent a precise comparison of the reached performances, an overall assessment of the previously discussed performances is here reported to compare the specific relevance of the computed space parameters. To do this, Figure 1 compares the reached AUC values as derived from the ROC curves corresponding to the best developed consensus models in all performed docking experiments and for both inhibition thresholds.

The analysis of the reported AUC values reveals results in substantial agreement with those previously discussed for EF1% values and allows for some considerations, which can be summarized as follows. The two explored spaces induce similar overall enhancing effects with the isomeric space, which appears to be more relevant for analyses with pIC50 > 5, reasonably due to the higher number of involved molecules existing in multiple states. The combinations of the two space parameters provide comparable performances, and rarely do they surpass those reached by the individual spaces. The LiGen program is that best benefitting from the inclusion of space parameters, while Fred is the most insensitive tool. When considering the best AUC values, Figure 1 reveals that the performances of the four docking programs are overall comparable for the screening campaigns with pIC50 > 5, while PLANTS yield lower AUC values with pIC50 > 6 compared to the other three pieces of software, which in turn afford rather similar performances.

Comparative analyses were also performed by calculating the corresponding consensus models using the common database. The obtained results (Appendix A) are in clear agreement with those previously discussed. The only difference involves the more limited enhancing role played by the isomeric space, which is ascribable to the reduced number of considered compounds existing in multiple states. The best consensus models developed using the common database will be used to compare the resulting rankings from the four tested docking programs (see below).

Figure 2 focuses on the enhancing role played by the development of the consensus models by showing the progressive effect exerted when including from two to five variables (in respect to the single scores). As already seen, the best improvements are reached by PLANTS and LiGen, while Glide and especially Fred show more limited effects. The beneficial role on the LiGen results might be also ascribed to the fact that their analyses involve the largest number of computed score values due to the inclusion of 30 pharmacophoric distances. However, a relation between performances and the number of variables cannot be evidenced for the other three docking programs.

When considering the progressive contribution when including from two to five variables in the consensus models, Figure 2 shows that the largest improvements are observed when shifting from one to two (around 14%) and from two to three variables (around 10%), while the inclusion of additional variables induces more limited EF1% increases (6% and 2% for four and five variables, respectively). On one hand, these results justify the choice made here of avoiding the calculation of consensus models with more than five variables. On the other hand, Figure 2 suggests that a simpler and faster analysis might be focused on consensus equations, including at most three variables that represent an optimal balance between performances, reliability, and computational costs.

To conclude this general analysis, Appendix A compiles the occurrence of the various scores as obtained by all the computed consensus models and reveals the major role played by the primary scores. Notably, primary scores also include the LiGen pharmacophoric distances, which alone represent more than 50% of all primary score values. Appendix A highlights the overall relevant role of both PLANTS and XScore scoring functions and consequently emphasizes the crucial role of rescoring procedures for enhancing the predictive power of all performed VS campaigns. The scores encoding for specific interactions play a minor role, even though Appendix A underlines the appreciable effect played by scoring functions encoding for non-polar interactions as seen by summing the relevance of both MLP and VEGA-based scores. Concerning score types, Appendix A confirms that best values represent about half of all included values and spread and mean scores show a more limited and similar incidence (around 25%).

Even though the analysis of the computed poses goes beyond the primary objective of the study, which was designed to evaluate the effects of the monitored space parameters in enhancing the performances of the performed VS campaigns, Figure 3 compares the best computed poses for a close TG-0205221 analogue included in the screened database with the recently resolved SARS-CoV-2 3CL-Pro complex [30]. While considering that the performed VS campaigns cannot simulate the formation of the covalent bond between Cys145 and the bound ligand, as seen in the reference structure (Figure 3A), the computed poses are in encouraging agreement with that of TG-0205221. Indeed, in all four shown structures, the 2-oxopyrrolidin ring approaches Asn142, the leucine side chain, which replaces the cyclohexyl alanine of TG-0205221 contacts His41, and the benzoyloxy moiety approaches Pro168 and Gln192. The four complexes slightly differ for the arrangement of the electrophilic warhead, even though it appears to be always close enough to Cys145 to yield the Michael adduct.

### 2.7. Mixed Consensus Models

The analyses on the common database was primarily carried out to develop mixed consensus equations by linearly combining scores coming from different docking simulations. For simplicity’s sake and considering the observed differences among the four docking runs even for the common database, the development of the mixed models was focused on the docking scores as such avoiding the space parameters.

Table 6 reports the best EF1% values as obtained by generating consensus equations that linearly combine scores of pairs of docking engines and the corresponding performance improvements (in percentage values). Overall, the reported EF1% values reveal interesting synergistic effects that affect most of the tested combinations with only four out of 12 cases showing no enhancement. On average, the synergistic enhancements are rather similar for both inhibition thresholds (i.e., 12% for pIC50 > 5 and 19% for pIC50 > 6). The best results are afforded by the combination of Glide plus Fred, which yield for both pIC50 thresholds EF1% better than those of the single docking program, while the best synergistic enhancement is seen when combining LiGen and PLANTS scores with pIC50 > 6 (40%).

Given these encouraging results, the next analysis involved the combination of triplets of programs. Table 6 also includes the EF1% values reached by these analyses and reveals an appreciable synergistic effect for these combinations with only one case being ineffective and most cases with an EF1% increase greater than 10%. Notably, these consensus models allow reaching EF1% around 20 for pIC50 > 5 and very close to 30 for pIC50 > 6. Unfortunately, the full combination of the scores coming from all tested docking programs did not yield further improvements (results not shown).

### 2.8. Analysis of the Best Rankings

The last section of this study analyzes the rankings obtained by applying the best consensus models for pIC50 > 6 using the common database (see Appendix A). The first part of the analysis investigates how the frequency of the molecules shared at the same time by two, three, or four rankings varies when browsing the first half of the ranking positions (Appendix A, from 1 to 5000). Appendix A shows the computed trends and reveals that the frequency of common molecules found in three rankings increases with a linear trend, while the frequencies of molecules included in all the four rankings or only in two rankings show symmetric and parabolic trends. The former grows when increasing the number of monitored ranking positions, and the latter symmetrically decreases.

Appendix A illustrates the parabolic trends as computed by considering specific pairs of rankings. This allows a graphical evaluation of the increasing overlapping between the results of two docking programs. Appendix A reveals that the highest frequency of shared compounds is provided by combining the LiGen and Fred rankings, while the other five pairs of programs yield rather similar profiles with the pairs LiGen–Plants and Plants–Fred showing the lowest frequencies of shared molecules. While being detectable even within the best top 100 ranking positions, the differences between the frequencies become appreciable when considering at least the first 500 ranking positions. Hence, the following analyses will focus on the molecules included in the top 500 ranking positions.

The first analysis of the top 500 molecules of each ranking deals with their physicochemical profiling. Table 7 reports the corresponding averages values and standard deviations for some key geometrical and physicochemical descriptors and allows for some relevant considerations. Firstly, limited differences are seen for the average values of the number of rotors and H-bonding groups, while molecular size (as encoded by M.W. and SAS averages) and polarity (as parameterized by PSA and log P averages) reveal a more marked variability. In detail, Table 7 shows that Glide and PLANTS select the bulkiest and the smallest set of ligands, respectively, while Fred and LiGen unravel intermediate averages. Additionally, there is an expected relation between size and lipophilicity for PLANTS, LiGen, and Fred, while Glide selects the ligands with a peculiar profile, since they comprise the bulkiest and the most polar molecules. Finally, concerning the property variability, Table 7 reports modest differences among the four monitored sets of ligands and for all computed descriptors, even though the Glide set shows, on average, the highest standard deviations, thus suggesting a conceivable relation between molecular size and property variability.

The next analysis on the top 500 ligands concerned the overlapping between the four sets of selected ligands. Figure 4 shows the resulting Venn diagram with the corresponding frequency values. As seen in Appendix A, Figure 4 confirms that the pairs Fred–LiGen and, to a minor extent, Glide–PLANTS show the highest frequencies of common molecules, while the pairs PLANTS–LiGen as well as PLANTS–Fred show the lowest degree of overlapping. Accordingly, the two triplets including Ligen and Fred (LiGen–Glide–Fred and PLANTS–LiGen–Fred) show the highest overlapping degree, while PLANTS–LiGen–Glide reveals the lowest number of shared ligands. Consequently, LiGen and Fred show the lowest numbers of unshared molecules, while PLANTS has the highest number of unique ligands, which roughly correspond to one half of the analyzed set. Finally, the molecules common to all the analyzed sets are 62, and this is a remarkable result, since they represent about 12.5% of the monitored ranking positions.

A similar diagram was also obtained when analyzing the common scaffolds, as detected within the screened Top 500 molecules (Appendix A). The scaffold frequencies are in excellent agreement with the ligand frequencies seen in Figure 4 (r^2^ = 0.97) and the total number of considered ligand frequencies (i.e., 1284) is slightly higher than that of scaffolds (i.e., 828). This means that each detected scaffold is shared on average by ≅1.5 ligands. Stated differently, these findings emphasize that the selected Top 500 molecules do not include congeneric series, and the chemical spaces covered by the top-ranked molecules of each docking software are rather similar (as also suggested by Table 7) with PLANTS and Glide showing the highest number of unshared scaffolds.

Finally, Appendix A compiles the common molecules shared by at least three rankings. In detail, the so collected common molecules are 194 (62 and 132 molecules shared by four and three rankings, respectively), among which 92 belong to the set of active compounds (with pIC50 > 5, i.e., 47%). This finding affords a further validation of the overall predictive power of the reported VS strategies, especially considering that the inhibition activity of several SARS-CoV 3CL-Pro inhibitors against the SARS-CoV-2 3CL-Pro enzyme was experimentally confirmed (as discussed above for TG-0205221 analogues. Among the other 102 molecules, there are 37 compounds that are known inhibitors of SARS-CoV 3CL-Pro but with pIC50 < 5, and 17 known inhibitors of the main proteases of other viruses (such as norovirus and HIV), and these compounds represent a further confirmation of the efficacy of the performed VS campaigns, since some of the retrieved hits (rupintrivir [31], saquinavir [32], and lopinavir [33]) were experimentally confirmed as promising inhibitors of SARS-CoV-2 3CL-Pro. Finally, among the other common molecules, cobicistat [34] and galloyl analogues [35] were identified as SARS-CoV-2 3CL-Pro inhibitors.

## 3. Methods

### 3.1. Library and Protein Structure Preparation

Virtual screening studies were performed on a repurposing library, containing a unique list of 13,057 drugs. They comprise the set of safe in man drugs, commercialized or under active development in clinical phases and retrieved from the Integrity database, plus the Fraunhofer’s BROAD Repurposing Library provided by Fraunhofer IME. The screened database also includes a set of 478 molecules, in particular preclinical compounds, identified as “CoV Inhibitors”, which were considered as the active training set in the reported optimization/validation analyses. Hence, the screened dataset overall included 13,535 unique molecules and the set of presumably active compounds was composed of 478 inhibitors with pIC50 > 5 of which 193 molecules show a pIC50 > 6.

Even though the database was collected by selecting safe in man molecules so that the obtained results could be used also for repurposing purposes, Appendix A compares some representative physicochemical properties of active and inactive, which show rather similar distributions. To better assess the reliability of the screened database and to appreciate the role of docking simulations, a ligand-based VS campaign involving about 100 structural and physicochemical descriptors was performed. The best EF1% values as obtained by the consensus models with five variables are very low (EF1% = 5.5 and 3.7 with pIC50 > 5 and pIC50 > 6, respectively). On one hand, these poor results confirm that the simulated database does not show significant differences between active and inactive ligands, which can bias the here presented docking results. On the other hand, the very modest performances reached by the ligand descriptors further emphasize the relevance of the here described structure-based VS workflows.

The reported docking simulations were performed by applying procedures as homogeneous as possible to render the obtained results as comparable as possible. Nevertheless, minor differences remain concerning the preparation of the 3D structures of both protein and ligands, primarily due to specific requirements of the docking software. Thus, the common procedures applied to prepare the input structures are here described, while the specific tasks required by each software will be reported in the following sections. All compounds were converted to 3D structures and prepared by using Schrödinger’s LigPrep tool. This process generated multiple states for stereoisomers, tautomers, ring conformations (one stable ring conformer by default), and protonation states. In particular, another Schrödinger package, Epik, was used to assign tautomers and protonation states that would be dominant at a selected pH range (pH = 7 ± 1) [36]. Ambiguous chiral centers were enumerated, allowing a maximum of 32 isomers to be produced from each input structure. Then, energy minimization was performed with the OPLS3 force [37]. In this way, 4515 compounds were characterized by multiple states (with an average of 3.1 states per compound), and a total of 23,654 ligands were generated. Docking simulations involved the monomer A of the first resolved SARS-CoV-2 3CL-Pro structure (PDB Id: 6LU7) in a covalent complex with the N3 inhibitor [9]. The protein structure preparation and the binding site characterization were performed as previously described [38]. Briefly, the protein structure was prepared by removing water solvents, crystallization additives, and the covalently bound N3 ligand. The hydrogen atoms were added by using the VEGA program [27] to remain compatible with physiological pH. The protein structure was then minimized using Namd2 [39] and by keeping the backbone atoms fixed to preserve the resolved folding.

### 3.2. PLANTS Simulations

Concerning PLANTS simulations, ligand conformations and atomic charges were further optimized by semi-empirical PM7 method as implemented by MOPAC [40]. Docking simulations were performed by PLANTS, which is based on ant colony optimization (ACO). [40] Docking search was focused within an 8 Å radius sphere around the co-crystallized N3 inhibitor and, for each compound, 10 poses were generated and ranked by the ChemPLP scoring function with the speed equal to 1. PLANTS and MOPAC calculations were carried out by exploiting Warpengine, an in house developed system for distributed computing [41]. For the post-docking analyses, all generated poses having the ChemPLP score > 0 were discarded, and this induced the loss of 75 inactive compounds (as seen in Table 1).

### 3.3. LiGen Simulations

The geometrical docking procedure implemented in LiGen™, proprietary software developed by Dompé, was used for the reported docking simulations [42]. In detail, the docking search was focused within a 5.0 Å radius sphere around the co-crystallized ligand. The available void volume of the resulting pocket was defined by determining the free points within a 3D grid, which encompass the entire binding site. The free points are used by the docking procedures as well as to define the pharmacophore schemes. Specifically, the docking engine follows a specific workflow during which three docking scores are computed: first, the Pacman Score (PS) estimates a geometric fitting by evaluating the interaction between a ligand pose and the pocket, based on shape and volume complementarity. Then, the Chemical Score (CS), which encodes for the ligand binding interaction energy, is calculated by an in-house developed scoring function [43]. The last step involves a rigid body minimization of the docked ligand within the binding site, at the end of which a third score called the optimized chemical score (Csopt) is evaluated. All poses that do not fulfill geometric fitting or thresholds values of user-defined specific parameters are discarded, and this induced the loss of 744 compounds (among which were only six active molecules, as seen in Table 1).

With regard to pharmacophore analysis, the program implements three different probe atoms, based on the Tripos Force Field, to explore the binding pocket: (1) a positively charged sp3 nitrogen atom (ammonium cation), describing a hydrogen bond donor; (2) a negatively charged sp2 oxygen atom (as in a carboxyl group), representing a hydrogen bond acceptor; and (3) an sp3 carbon atom (methane), encoding for a hydrophobic group. The representative atom types can be modified, even though this selection produced the best outcomes in previous benchmarking analyses. For each free grid point, the binding energies between the probes and the protein atoms are evaluated by using an in house extended scoring function based on the work of Wang [39]. Every grid point will be identified as donor, acceptor, or hydrophobic according to which probe yields the best score.

The software then filters all the grid points to extract the key interaction sites in three steps. Firstly, the program averages the scores of all the grid points for each probe and selected those points having a score lower than the average. Based on these favorable grid points, LiGenPocket finds the pharmacophoric features. They are identified by clustering the neighbors’ grid points, which are here defined as grid points with the same definition (donor, acceptor, or hydrophobic), falling at a distance less than 2.0 Å from a given point. The score averages of all points belonging to the so identified clusters are computed for each type of grid points, and those points, which show a score value lower than the average of their cluster, are discarded.

Secondly, the clusters of neighbor grid points that survive to the previous filtering process constitute the pharmacophoric features of the binding site. The geometric center of each cluster is thus defined as a pharmacophoric point. Lastly and for each pharmacophore element, the minimum distances between a ligand’s atom and the closest compatible pharmacophoric point are calculated for each ligand. Each atom of the ligand is defined by a classification that parallels that of the used probes: four atom types are indeed considered and defined by a letter code (A, D, AD, and H are Acceptor, Donor, Acceptor and Donor, and Hydrophobic, respectively).

### 3.4. FRED Simulations

Ligand conformers were generated using OpenEye OMEGA [44]. Conformers with internal clashes and duplicates were discarded by the software, and the remaining ones were clustered based on of the root mean square deviation (RMSD). For this virtual screening, a maximum of 200 conformers per compound, clustered with an RMSD of 0.5 Å, was used. If the number of conformers generated exceeds the specified maximum, only the ones with the lowest energies are retained. For 1780 molecules, the generation of the rotamers was not possible due to stereochemistry issues and/or for the presence of large macrocycles, and they were removed from the library. The resulting library consisted of 11,755 molecules. The target protein was processed using UCSF Chimera (v1.14). AMBER ff14SB was used to assign parameters to the standard residues, whereas the Antechamber module was used for the nonstandard residues. The charges for the nonstandard residues were calculated using the AM1-BCC method. The structure was minimized with 100 steps of gradient descent and 10 steps of conjugate descent, using the MMTK module. Rigid docking was then performed using OpenEye FRED [45] included in the OEDocking 3.4.0.2 suite (OpenEye Scientific Software, Santa Fe, NM. http://www.eyesopen.com). Each docked pose is scored using the Gaussian Shape scoring function. Finally, top scoring poses are converted into density fields to form the final shape potential field. The highest values in this field represent points where molecules can have a high number of contacts, without clashing into the protein structure. In its exhaustive search, FRED translates and rotates the structure of each conformer within the negative image of the active site and scores each pose. FRED first step has a default translational and rotational resolution of 1.0 and 1.5 Å, respectively. The 100 best scoring poses are then optimized with translational and rotational single steps of 0.5 and 0.75 Å, respectively, exploring all the 729 (six degrees of freedom with three positions = 36) nearby poses. The best scoring pose is retained and assigned to the compound. The binding poses were evaluated by using the Chemgauss4 scoring function implemented in OpenEye FRED [28]. For 24 molecules, the docking algorithm was unable to find a suitable binding pose, and these molecules were thus discarded from the analysis.

### 3.5. Glide Simulations

To perform docking experiments with Glide, the protein was preprocessed by the Protein Preparation Wizard from the Schrodinger Suite version 2019-4 with the default parameters [28]. The protonation states of each side chain were generated using Epik for pH = 7 ± 2 [36]. Protein minimization was performed using the OPLS3 force field [37]. All water molecules were removed. Glide software [29] was used for the docking calculations. Internal receptor grid boxes of 10 Å × 10 Å × 10 Å were defined and centered on the ligand atom position. The size of the outer binding box was determined by the ligand size (27 × 27 × 27 Å). A standard precision (SP) Glide docking was carried out, generating 20 poses per docked molecule. H-Bond constraints with D166, H163, and H164 were applied. Docking results were analyzed by Glide Docking score in the version 5.0 [46]. Here, Glide score was used to extract the best binding pose for each ligand. This is an empirical scoring function able to reproduce the trends of the binding affinity and is defined by the following equation:GScore=a.vdW+b.Coul+Lipo+Hbond+Metal+Rewards+RotB+Site
where: *vdW* = van der Waals interaction energy; *Coul* = Coulomb interaction energy; *Lipo* = Lipophilic-contact plus phobic-attractive term; *HBond* = Hydrogen-bonding term; *Metal* = Metal-binding term (usually a reward); *Rewards* = Various reward or penalty terms; *RotB* = Penalty for freezing rotatable bonds; *Site* = Polar interactions in the active site, and the coefficients of vdW and Coul are: *a* = 0.050, *b* = 0.150 for Glide 5.0 (the contribution from the Coulomb term is capped at −4 kcal/mol).

### 3.6. Rescoring Calculations

All the poses generated by the four docking programs were rescored by ReScore+ [47]. The computed scoring functions comprise (a) the various components of PLANTS [25] and XScore [24] scoring functions; (b) a set of scores computed by the VEGA suite, which encodes for polar and non-polar interaction energies [27]; (c) the MLP interactions scores for hydrophobic contacts [26]; (d) the recently proposed Contacts scores [18], which are simply based on several surrounding residues, and (e) the APBS score for evaluating ionic interactions [48]. Both the primary scores and the values from rescoring calculations were utilized to calculate binding and isomeric spaces as well as their combinations by applying a joining and a merging strategy (see below). For each considered scoring function, each explored space is defined by the following values: (1) the best scores including both the lowest and the highest values (notice that the best value is not the lowest one for all scores); (2) the average score value; and (3) the score range and the standard deviation to encode for the spread of score values. For each ligand, all the generated poses were utilized to calculate the corresponding space parameters without exceptions.

In detail, the binding space was computed by averaging the computed scores for the poses of a given molecule/isomer. For molecules existing in multiple states, the space parameters corresponding to the isomer with the best primary score were considered. Similarly, the isomeric space was calculated by averaging the computed scores of the pose with the best primary score for all the isomers (clearly only for molecules existing in multiple states). In the so-called merged combination, the space parameters were calculated by averaging together the computed scores of all poses and all isomers. In the so-called joint combination, the consensus equations were developed by simultaneously considering the space parameters as computed for both binding and isomeric spaces. The descriptors for the binding and isomeric spaces were computed by using ad-hoc scripts of the VEGA suite of programs [27].

### 3.7. Consensus Analyses

The consensus analyses involved the primary scores and the scoring functions as computed by rescoring procedures. Notably, the analysis of the LiGen results also comprised the pharmacophoric distances as computed by this tool. The consensus analyses were performed by the EFO approach, which generates linear combinations of score values by exhaustively combining all possible variables and by optimizing a quality function based on both the early recognition (as encoded by the corresponding EF 1% values) and the entire ranking (as encoded by an asymmetry index applied to the distribution of the active molecules) [33].

By considering the high number of here analyzed descriptors along with the already included exhaustive search method, an incremental search algorithm was also implemented. In particular, given n descriptors in the input dataset, the equations with k variables are built by considering only the top ranked m equations with k − 1 variables (m is a user-defined parameter and is set by default equal to 30) and by combining them with the n descriptors avoiding repetitions. Therefore, the models to be evaluated are m (n − k + 1) instead of the number of all possible combinations without repetitions, which are equal to n!/k! (n − k)!. A benchmark analysis by comparing the consensus models as generated by incremental and by exhaustive searches revealed that the former involves a performance decrease of about 10 % as assessed by the relative EF1% values when generating the three variable equations for the simplest case without space parameters (data not shown). Such a performance loss is seen as acceptable by considering that the incremental search algorithm allows the analysis of very extended datasets of descriptors and the development of consensus models including more than three variables. Furthermore, the reduced performances of the incremental algorithm equally affected all the here performed comparative analyses. This new search approach was implemented into a standalone and highly optimized version of EFO, which does not require the full installation of the VEGA program and can be freely downloaded at www.vegazz.net. In detail, 20 consensus models were generated for each analysis by combining all input variables without preliminary filtering processes and by always using the incremental search approach. The consensus equations were developed by including from one to five variables. The predictive power of the resulting equations was assessed by subdividing the dataset into training (70%) and test sets (30%) and repeating this task 10 times to minimize the randomness.

## 4. Conclusions

The study describes and compares a set of VS campaigns performed by using four different docking tools to repurpose an extended set of safe in man molecules as potential inhibitors of the SARS-CoV2 3CL-Pro enzyme. To assess the predictive performances of the here proposed docking strategies and due to the lack of known inhibitors for the considered enzyme, the peculiar idea of the study is the exploitation of a training set composed of ≅500 compounds that were reported as effective inhibitors (i.e., pIC50 > 5) of the SARS-CoV 3CL-Pro enzyme, a choice justified by the very high conservation degree between these two viral proteases.

All docking simulations were carried out by generating more than one pose per ligand and explicitly considering all possible isomers/states for those molecules existing in multiple states. In this way and after a rescoring analysis of all computed poses, the obtained results were utilized to calculate the descriptors of the resulting binding and isomeric space. The so-calculated score values for each utilized docking program were finally employed to develop consensus models by linearly combining them using the EFO approach.

Taken together, the obtained results allow for some concluding considerations, which can be summarized as follows:(a)Apart from Fred, the performances of all other docking programs benefit from the inclusion of the binding space parameters as witnessed by an overall EF1% increase average equal to 18% with pIC50 > 6.(b)The inclusion of the isomeric space parameters reveals, on average, a poorer beneficial effect and plays a clear role only with pIC50 > 5, a finding explainable considering the low number of actives existing in multiple states with pIC50 > 6.(c)The merging combination of the two explored spaces is substantially ineffective, while the joint combination provides relevant enhancing effects, especially when analyzing the PLANTS and Glide results.(d)The beneficial effects of the inclusion of space parameters markedly vary among the utilized docking engines; specifically, PLANTS and Glide appear to benefit from both spaces, the LiGen performances are positively affected only by the binding space, while Fred appears to be substantially insensitive to both spaces.(e)Even though the primary scores play a relevant role in almost all generated predictive models, the rescoring procedures reveal a remarkably beneficial impact by increasing the performances of all used programs regardless of their sensitivity to space parameters.(f)The enhancing effect of linearly combining diverse score values in the consensus models also varies among the docking programs with PLANTS and Fred showing the largest and the poorest overall effect, respectively.(g)Although the primary objective of the study was to assess the role played by various post-docking procedures and the retrieved hits were not experimentally tested, it should be noted that the best ranked compounds include an encouraging number of heterogeneous molecules, the SARS-CoV-2 3CL-Pro inhibition activity of which was recently reported (see Appendix A). This represents an indirect confirmation of the reliability of the reported simulations.

Since the SARS-CoV2 3CL-Pro is functionally active as a homodimer and many X-ray structures were recently resolved, the encouraging results here reported invite to repeat similar docking protocols by simultaneously considering both the monomers of different representative resolved structures to evaluate the enhancing effects exerted by binding and isomeric spaces for the resulting ensemble simulations. Not to mention that the same strategies could be also applied to explore representative frames coming from MD simulations.

Even though the employment of the known SARS-CoV 3CL-Pro inhibitors was justified by the very high conservation degree between these two enzymes, there is no doubt that the developed predictive models could be enhanced by utilizing true SARS-CoV 3CL-Pro inhibitors. One may figure out that the here described simulations could be exploited in a near future to develop increasingly performing predictive models as novel true inhibitors are identified.

To conclude, the here reported VS campaigns emphasize the generally beneficial effects of the applied post-docking procedures, even though their specific roles significantly vary among the utilized pieces of software. These observed differences can be ascribed to the different implemented algorithms for docking search and scoring calculations, but they can also be due to the intrinsic geometrical and physicochemical features of the SARS-CoV-2 3CL-Pro binding pocket. This last consideration emphasizes that a robust assessment of the role of binding and isomeric spaces could require extensive benchmarking studies involving wide sets of diverse target proteins.

## Figures and Tables

**Figure 1 molecules-26-00797-f001:**
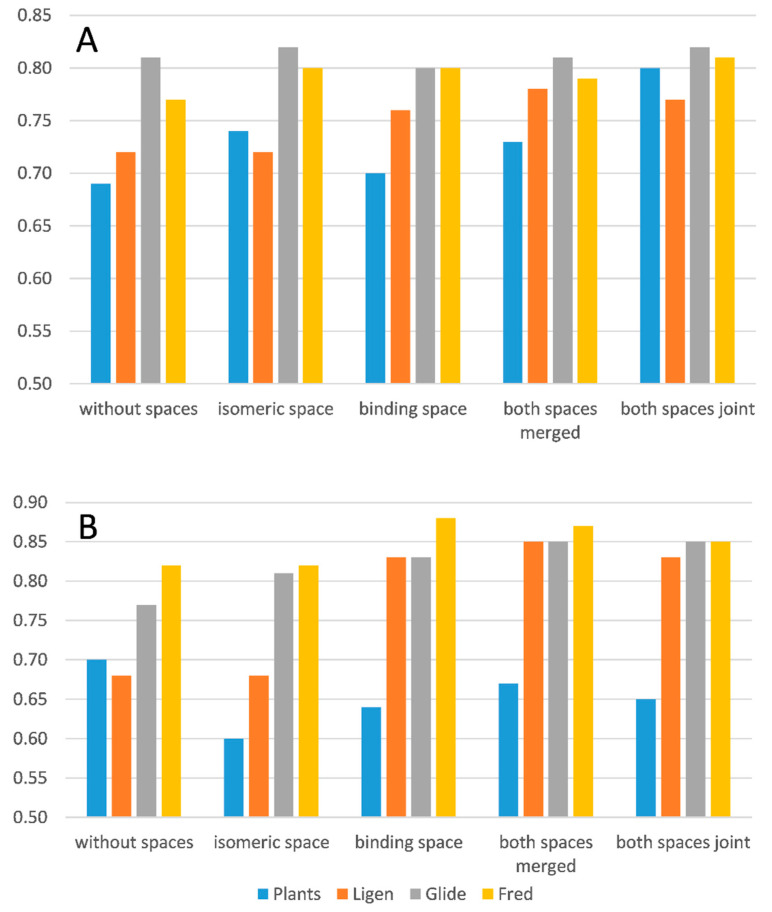
Comparison of the AUC values from the ROC curves for the best performing models generated by the four utilized docking programs in the five tested conditions with (**A**) pIC50 > 5 and (**B**) pIC50 > 6.

**Figure 2 molecules-26-00797-f002:**
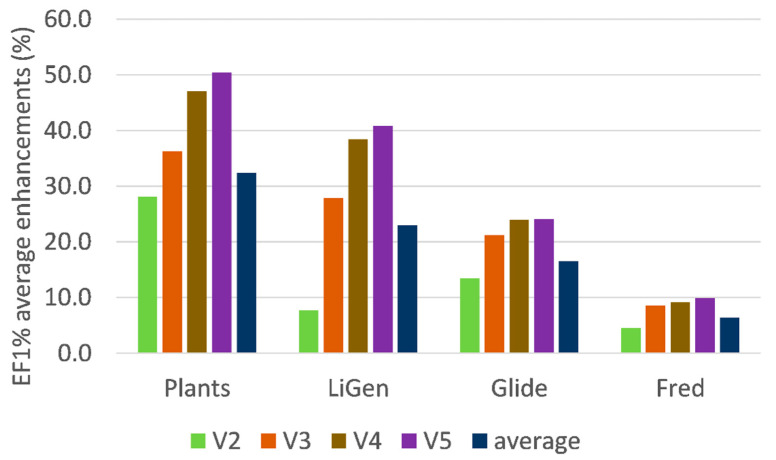
Average EF1% enhancements (in percentage values) due to the generation of the consensus models when including two (V2), three (V3), four (V4), or five (V5) variables plus overall averages. The reported values are computed by averaging the EF1% enhancements for both inhibition thresholds.

**Figure 3 molecules-26-00797-f003:**
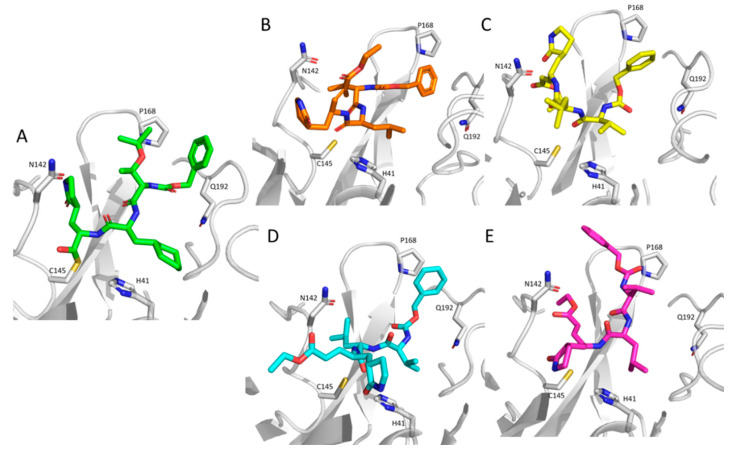
Comparison of the resolved complex of TG-0205221 with the SARS-CoV-2 3CL Pro enzyme (**A**) with the best poses as computed for its close analogue by PLANTS (**B**), LiGen (**C**), Fred (**D**), and Glide (**E**).

**Figure 4 molecules-26-00797-f004:**
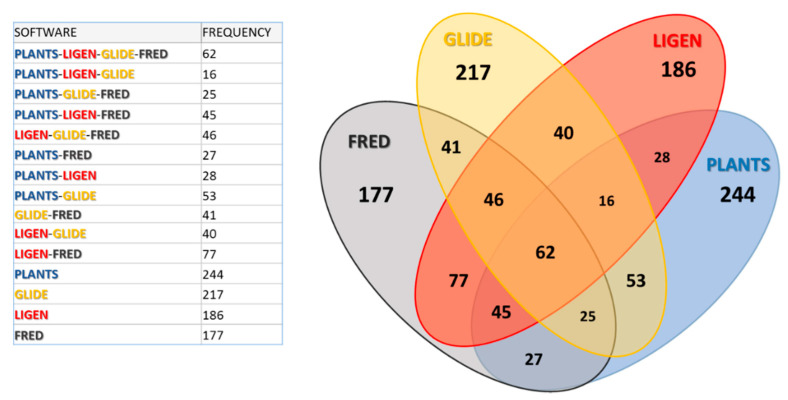
Venn diagram representing the overlapping degree between the Top 500 molecules as derived by applying the best equations developed for each docking software. A table detailing the resulting frequencies is also included (the color code for the four docking tools is the same adopted in Figure 1).

**Table 1 molecules-26-00797-t001:** General characteristics of the databases simulated by each docking program and relative generated poses.

Tool	Docked Molecules	pIC50 > 6	pIC50 > 5	Molecules with Multiple States	Monitored Poses
Full Database	13,535	193	478	4548	---
Glide	11,583	164	415	3470	174,071
Fred	11,731	138	411	3379	183,653
Ligen	12,791	190	472	3907	128,133
Plants	13,460	193	478	4475	224,670
Common	10,110	114	353	--- ^a^	--- ^a^

**^a^** The number of both molecules existing in multiple states and generated poses varies among the docking programs.

**Table 2 molecules-26-00797-t002:** Best EF1% values reached by the various consensus models developed using the PLANTS results plus the relative average values and the corresponding performance enhancements in percentage values. As described under Methods, the consensus equations were generated by linearly combining from two to five docking scores. The EF1% values referring to one variable correspond to the performances reached by single scoring functions.

Number of Variables	Without Spaces	Isomeric Space	Binding Space	Both Spaces Merged	Both Spaces Joint	Mean
**pIC50 > 5**						
**1**	7.1	9.0 (27%) ^a^	9.0 (27%) ^a)^	10.1 (41%) ^a^	9.0 (27%) ^a^	8.9
**2**	10.3	10.3 (0%)	10.3 (0%)	11.3 (10%)	10.3 (0%)	10.5 (18%) ^b^
**3**	10.3	11.8 (14%)	10.3 (0%)	11.3 (10%)	11.3 (15%)	11.1 (24%) ^b^
**4**	10.7	11.8 (10%)	10.7 (0%)	12.2 (14%)	11.6 (10%)	11.4 (28%) ^b^
**5**	10.7	11.8 (10%)	11.6 (8%)	13.0 (22%)	12.0 (12%)	11.8 (33%) ^b^
**Mean**	9.8	10.9 (11%)	10.4 (5%)	11.6 (18%)	11.0 (12%)	10.7 (20%) ^b^
**pIC50 > 6**						
**1**	9.4	10.9 (16%)	9.4 (0%)	9.4 (0%)	10.9 (16%)	10.0
**2**	13.4	14.1 (5%)	13.4 (0%)	14.6 (8%)	14.1 (5%)	13.8 (38%)
**3**	13.4	15.1 (12%)	13.4 (0%)	15.6 (16%)	16.7 (24%)	14.8 (48%)
**4**	14.1	15.1 (7%)	18.2 (29%)	17.2 (21%)	18.2 (29%)	16.6 (66%)
**5**	14.1	15.1 (7%)	18.2 (29%)	18.2 (29%)	18.2 (29%)	16.8 (68%)
**Mean**	12.9	14.6 (9%)	15.3 (13%)	15.9 (16%)	16.0 (20%)	14.4 (43%)

^a^ Here and in the following rows, the performance enhancements are computed in respect to the first column reporting the EF1% values obtained without including space parameters to assess the beneficial role of the inclusion of these parameters and their combinations. The same holds for pIC50 > 6 and Tables 4, 6, and 8. ^b^ In the last column, the enhancements are computed with respect to the average value obtained by the single scoring functions to evaluate the average effect of the consensus models. The same holds for pIC50 > 6 and Tables 4, 6, and 8.

**Table 3 molecules-26-00797-t003:** Best EF1% values reached by the various consensus models developed using the LiGen results plus the relative average values and the corresponding performance enhancements in percentage values. As described under Methods, the consensus equations were generated by linearly combining from two to five docking scores. The EF1% values referring to one variable correspond to the performances of single scoring functions.

Number of Variables	Without Spaces	Isomeric Space	Binding Space	Both Spaces Merged	Both Spaces Joint	Mean
**pIC50 > 5**						
**1**	11.3	11.3 (0%)	11.3 (0%)	11.3 (0%)	11.3 (0%)	11.3
**2**	11.3	12.8 (13%)	12.4 (9%)	11.3 (0%)	12.8 (9%)	12.0 (7%)
**3**	13.9	13.9 (0%)	14.5 (5%)	13.9 (0%)	14.5 (7%)	14.1 (25%)
**4**	14.5	14.5 (0%)	16.2 (12%)	14.9 (3 %)	16.2 (12%)	15.3 (34%)
**5**	14.5	15.1 (4%)	16.2 (12%)	15.6 (7%)	16.2 (12%)	15.5 (36%)
**Mean**	13.1	13.5 (3%)	14.1 (8%)	13.4 (2%)	14.2 (8%)	13.8 (20%)
**5 no PH ^a^** **Distances**	9.4	9.4	9.4	9.4	11.11	9.7
**5 only PH** **Distances ^a^**	14.5	15.1	13.0	8.5	15.6	13.3
**pIC50 > 6**						
**1**	12.7	12.7 (0%)	12.7 (0%)	12.7 (0%)	12.7 (0%)	12.7
**2**	13.2	13.2 (0%)	14.8 (12%)	14.3 (8%)	14.8 (12%)	13.8 (8%)
**3**	13.2	13.8 (4%)	21.2 (60%)	21.2 (60%)	21.2 (60%)	16.6 (31%)
**4**	13.2	13.8 (4%)	24.9 (88%)	25.4 (92%)	24.9 (88%)	18.2 (43%)
**5**	13.8	13.8 (0%)	25.9 (89%)	25.4 (84%)	25.9 (89%)	18.5 (46%)
**Mean**	13.2	13.4 (2%)	19.9 (50%)	19.8 (49%)	19.9 (50%)	16.0 (26%)
**5 no PH** **Distances ^a^**	10.6	11.6	12.2	21.3	20.1	15.2
**5 only PH** **Distances ^a^**	9.0	11.1	10.6	12.2	11.1	10.8

**^a^** In both analyses, the last two rows report the performances reached by the consensus equations generated by combining five variables and excluding the pharmacophoric distances (“five no PH distances”) or considering only these parameters (“five only PH distances”).

**Table 4 molecules-26-00797-t004:** Best EF1% values reached by the various consensus models developed using the Fred results plus the relative average values and the corresponding performance enhancements in percentage values. As described under Methods, the consensus equations were generated by linearly combining from two to five docking scores. The EF1% values referring to one variable correspond to the performances of single scoring functions.

Number of Variables	Without Spaces	Isomeric Space	Binding Space	Both Spaces Merged	Both Space Joint	Mean
**pIC50 > 5**						
**1**	13.7	14.9 (9%)	15.1 (11%)	16.6 (21%)	15.1 (11%)	15.1
**2**	14.9	15.1 (2%)	16.3 (10%)	16.6 (11%)	17.1 (15%)	16.0 (6%)
**3**	16.1	16.3 (1%)	16.6 (3%)	17.3 (8%)	17.6 (9%)	16.8 (11%)
**4**	16.1	16.3 (1%)	16.6 (3%)	17.3 (8%)	17.6 (9%)	16.8 (11%)
**5**	16.1	16.3 (1%)	16.6 (5%)	17.3 (8%)	17.6 (9%)	16.8 (11%)
**Mean**	15.4	15.8 (3%)	16.3 (6%)	17.0 (11%)	17.0 (11%)	16.3 (8%)
**pIC50 > 6**						
**1**	24.7	24.7 (0%)	24.7 (0%)	24.7 (0%)	24.7 (0%)	24.7
**2**	25.4	25.4 (0%)	25.4 (0%)	25.4 (0%)	25.4 (0%)	25.4 (3%)
**3**	25.4	25.4 (0%)	25.4 (0%)	26.9 (6%)	27.6 (9%)	26.2 (6%)
**4**	25.4	25.4 (0%)	26.9 (6%)	26.9 (6%)	27.6 (9%)	26.4 (7%)
**5**	25.4	25.4 (0%)	27.6 (9%)	27.6 (9%)	27.6 (9%)	26.7 (8%)
**Mean**	25.3	25.3 (0%)	26.0 (3%)	26.3 (4%)	26.6 (5%)	25.9 (5%)

**Table 5 molecules-26-00797-t005:** Best EF1% values reached by the various consensus models developed using the Glide results plus the relative average values and the corresponding performance enhancements in percentage values. As described under Methods, the consensus equations were generated by linearly combining from two to five docking scores. The EF1% values referring to one variable correspond to the performances reached by single scoring functions.

Number of Variables	Without Spaces	Isomeric Space	Binding Space	Both Spaces Merged	Both Spaces Joint	Mean
**pIC50 > 5**						
**1**	12.9	13.8 (8%)	13.1 (2%)	12.9 (0%)	13.8 (8%)	13.3
**2**	13.8	15.3 (11%)	14.3 (4%)	14.3 (4%)	16.5 (19%)	14.9 (12%)
**3**	14.3	16.0 (12%)	16.0 (12%)	16.3 (14%)	16.5 (15%)	15.8 (19%)
**4**	14.3	16.8 (17%)	16.3 (14%)	16.3 (14%)	17.2 (20%)	16.2 (22%)
**5**	14.3	16.8 (17%)	16.3 (14%)	16.3 (14%)	17.5 (22%)	16.2 (22%)
**Mean**	13.9	15.7 (13%)	15.2 (9%)	15.2 (9%)	16.3 (17%)	15.3 (15%)
**pIC50 > 6**						
**1**	19.7	22.7 (16%)	20.3 (3%)	19.7 (0%)	22.7 (16%)	21
**2**	22.1	25.8 (17%)	24.0 (8%)	23.3 (6%)	25.8 (17%)	24.2 (15%)
**3**	22.7	27.6 (22%)	25.8 (14%)	25.2 (11%)	28.3 (24%)	25.9 (23%)
**4**	24.0	27.6 (15%)	27.0 (13%)	25.8 (8%)	28.3 (18%)	26.5 (26%)
**5**	24.0	27.6 (15%)	27.0 (13%)	25.8 (8%)	28.3 (18%)	26.5 (26%)
**Mean**	22.5	26.3 (17%)	24.8 (10%)	24.0 (8%)	26.7 (19%)	24.8 (18%)

**Table 6 molecules-26-00797-t006:** Best EF1% values as obtained by combining the simple score values (without space parameters) of two or three different docking programs and the corresponding synergistic effect (as enhancements in percentage values). For easy comparison and concerning the results for pairs of docking tools, the diagonal cells reports the best EF1 value obtained by the single docking tool (in italics).

Program	PLANTS	LiGen	Fred	Glide
**Results for Pairs of Programs**				
**pIC50 > 5**				
**PLANTS**	*10.7*	15.3 (6%)	16.1 (0%)	16.2 (13%)
**LiGen**		*14.5*	16.1 (0%)	16.7 (15%)
**Fred**			*16.1*	18.1 (12%)
**Glide**				*14.3*
**pIC50 > 6**				
**PLANTS**	*14.1*	19.8 (40%)	28.1 (11%)	24.0 (0%)
**LiGen**		*13.8*	27.7 (9%)	24.0 (0%)
**Fred**			*25.4*	29.0 (16%)
**Glide**				*24*
**Results for triplet of programs**				
	Fred Glide PLANTS	LiGen Glide PLANTS	LiGen Glide Fred	LiGen Fred PLANTS
**pIC50 > 5**	18.1 (12%)	17.1 (18%)	18.7 (16%)	16.4 (2%)
**pIC50 > 6**	29.9 (18%)	24 (0%)	27.2 (7%)	28.1 (11%)

**Table 7 molecules-26-00797-t007:** Average values plus standard deviations for some key geometrical and physicochemical properties as computed by considering the Top 500 molecules of each ranking.

Tool	Rotors	HB_Tot	M.W.	PSA	SAS	Log P
**Plants**	9.6 ± 3.8	9.6 ± 2.2	510.9 ± 98.2	131.6 ± 45.2	541.8 ± 111.3	2.3 ± 2.7
**LiGen**	10.2 ± 4.2	9.2 ± 2.9	554.2 ± 95.6	120.8 ± 54.3	593.0 ± 98.3	3.6 ± 2.2
**Fred**	9.9 ± 4.4	9.2 ± 2.9	545.0 ± 97.8	120.3 ± 56.4	579.9 ± 104.6	2.9 ± 2.4
**Glide**	10.9 ± 4.2	11.1 ± 3.1	566.0 ± 103.1	148.3 ± 58.9	590.6 ± 109.4	2.2 ± 2.9

## Data Availability

Data available in a publicly accessible repository that does not issue DOIs. This data can be found here: http://www.exscalate4cov.network.

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
