# Peer review of "Combining Different Docking Engines and Consensus Strategies to Design and Validate Optimized Virtual Screening Protocols for the SARS-CoV-2 3CL Protease"

_molecules, 2021, doi:10.3390/molecules26040797_

Round 1

Reviewer 1 Report

The present manuscript tested the validity of four common docking software for identifying potential inhibitors of SARS-CoV-2 3CL protease. Moreover, consensus models were also developed by combining different docking engines, scoring functions, and other descriptors. The study is timely and provides useful strategies to fight against coronavirus in a long term. The manuscript is reasonably written, and the conclusions are well supported by the achieved results. However, the manuscript still needs a certain level of improvement before publication. After a careful revision, I suggest publishing their manuscript on Molecules.

Major comment:

1) The authors borrowed the inhibitors targeting SARS-CoV’s 3CL protease for validating the performance of different software on SARS-CoV-2’s 3CL protease. They assume that the two homologs proteases mostly share the existing inhibitors. I understand that the authors do not have better options to have sufficient confirmed inhibitors for constructing the test sets for SARS-CoV-2’s 3CL protease, but they still could rationalize this assumption by more analyses. I suggest two things:

A. Compare sequence and structural similarities between the two proteases and show how the inhibitors interact with the proteases (maybe use a crystal complex structure of SARS-CoV’s 3CL protease) by some representative inhibitors. It is unreasonable for a docking paper not to show any interaction analysis.

B. Because the authors used SARS-CoV’s inhibitors as the surrogates for Cov-2, why not validate the docking performance and new protocols on the real target protein of the collected inhibitors (SARS-CoV)? The author may not have to repeat all the simulations as they already did, but at least some key validations, for example, VS performance of each software.

2) I do not suggest making a strong conclusion about the software comparison. Based on the author’s descriptions for the setup for each software, the four software were not running at the same condition. For example, four software used different receptor grid box sizes for space exploration. If the comparison is not fair, simply telling which software is better may be quite misleading. I guess the authors mostly used the default setting (suggested by the developers) for the docking. If so, it is reasonable for a general docking simulation, but not for a fair comparison. Comparison between different docking software had been extensively studied before. I think the main purpose of the manuscript was to develop a new protocol for screening the hot target rather than fair comparison. Thus, I suggest weakening the comparison and focus on the combination of advantages from different docking engines. The authors still need to analyze the VS performance but do not make a strong conclusion about the comparison.

3) Why did the authors used the enrich factor (EF) as the main indicator for the VS evaluation? As I know, the ROC curve and its related metrics (e.g., AUC) are better for this purpose. It is much less dependent on the size difference of screened libraries. As the authors found, four docking software failed to dock some compounds, evaluating VS performance with EF may not suitable for resulting docked libraries with varied numbers of compounds.

4) The authors should also evaluate the accuracy of pose reproducibility. Of course, such evaluation will mainly use SARS-CoV’s 3CL and their inhibitors but may also include some SARS-CoV-2 3CL’s (if any). The authors may collect as many crystal complexes as possible to do this. Although evaluating the VS performance is the key in the manuscript, how useful each software for the pose prediction will be also very important information for further optimization of hit compounds.

Minor comment:

1) Physio-chemical properties should be analyzed by providing some statistical figures like MW, logP, and PSA. Also, some representative chemical structures (2D) should be presented in the main text.

2) The authors used too many tables in the manuscript. They could be easily replaced by some well-organized statistical plots. Moreover, please give clear descriptions for each notation in tables or plots. I understand the author may describe them somewhere in the main text, but it is difficult to follow the tables without clear notations.

3) How did the authors prepare the protein structures? in “3.1 Library and protein structure preparation”, the author did not give a general procedure. Instead, they described the protein preparation procedures separately for different software. I understand different docking software may have some special requirements, but the protonation states of the key residues in the binding site should be the same for different docking software. The authors should carefully clarify the protein preparation procedure. I suggest giving a general procedure first and then the specialized ones for each software.

4) In the “method” section, the sequential number of “3.3 PLANTS simulations” is not correct.

5) “3.6 Rescoring and consensus analyses” is very lengthy and not well organized. The authors should divide it into several paragraphs and put some flowcharts to facilitate understanding.

6) How did the authors disperse 478 inhibitors into a database of ~13000? What does “13000 purposely collected safe in man molecules” mean? I think the authors should also provide the statistical plots for physicochemical properties of these decoy molecules as I suggested for the inhibitors. Besides, why did not the authors construct a DUD-like decoy library for the VS evaluation?

7) For lines 118 to 119: how do the score averages account for the dynamic profile of the binding process, and the range values encode for the often-neglected entropic component? Could the authors provide references to support their statements? But anyway, I do not think docking scores can give a good estimation for the pose dynamics and binding entropies.

8) For lines 169 to 171: I agree with “the common database” is useful to develop a mixed consensus model. But I do not understand why it is also important for “carefully compare the performances of each docking program”. According to my opinion, failed docked molecules for a specific docking software indicate that the software and/or scoring functions do not “like” those molecules, and those failed molecules can be just considered as the ones with bad docking scores. Those compounds can be still used in the VS evaluation by putting them at the end of the hit list.

9) For line 194: should the number 479 be corrected to 478?

10) Please give more clear descriptions for “isomeric space”, “binding space”, “both spaces merged”, “both spaces joint”. Give more details about the ways of their constructions. If necessary, please write down the representative equations.

11) Give the details about “n. variables” (what does 1, 2, 3, 4, and 5 mean) together with tables or figures.

Author Response

Dear Reviewer,

We thank you for your valuable suggestions. Here is a description of the amendments made in the revised version according to your requests.

The present manuscript tested the validity of four common docking software for identifying potential inhibitors of SARS-CoV-2 3CL protease. Moreover, consensus models were also developed by combining different docking engines, scoring functions, and other descriptors. The study is timely and provides useful strategies to fight against coronavirus in a long term. The manuscript is reasonably written, and the conclusions are well supported by the achieved results. However, the manuscript still needs a certain level of improvement before publication. After a careful revision, I suggest publishing their manuscript on Molecules.

Major comment:

1) The authors borrowed the inhibitors targeting SARS-CoV’s 3CL protease for validating the performance of different software on SARS-CoV-2’s 3CL protease. They assume that the two homologs proteases mostly share the existing inhibitors. I understand that the authors do not have better options to have sufficient confirmed inhibitors for constructing the test sets for SARS-CoV-2’s 3CL protease, but they still could rationalize this assumption by more analyses. I suggest two things:

A. Compare sequence and structural similarities between the two proteases and show how the inhibitors interact with the proteases (maybe use a crystal complex structure of SARS-CoV’s 3CL protease) by some representative inhibitors. It is unreasonable for a docking paper not to show any interaction analysis.

A paragraph better detailing the similarity between the two proteases was added in the Introduction. Moreover, Figure 3 compares the poses for the TG-0205221 inhibitor in the resolved complexes for the two enzymes.

B. Because the authors used SARS-CoV’s inhibitors as the surrogates for Cov-2, why not validate the docking performance and new protocols on the real target protein of the collected inhibitors (SARS-CoV)? The author may not have to repeat all the simulations as they already did, but at least some key validations, for example, VS performance of each software.

We thank the reviewer for the suggestion. However, we think that the study is already very rich in the analysis of the performances of docking simulations. Therefore, we do not believe that the proposed simulations could further justify our assumption. Especially considering the obtained results would yet depend on the chosen SARS-CoV’s 3CL protease resolved structure and the adopted settings. We think that the similarities added at point A are enough to rationalize the employment of the SARS-CoV’s 3CL protease inhibitors as active molecules which, however, remains an approximation.

2) I do not suggest making a strong conclusion about the software comparison. Based on the author’s descriptions for the setup for each software, the four software were not running at the same condition. For example, four software used different receptor grid box sizes for space exploration. If the comparison is not fair, simply telling which software is better may be quite misleading. I guess the authors mostly used the default setting (suggested by the developers) for the docking. If so, it is reasonable for a general docking simulation, but not for a fair comparison. Comparison between different docking software had been extensively studied before. I think the main purpose of the manuscript was to develop a new protocol for screening the hot target rather than fair comparison. Thus, I suggest weakening the comparison and focus on the combination of advantages from different docking engines. The authors still need to analyze the VS performance but do not make a strong conclusion about the comparison.

As suggested by the reviewer, all the comparisons on the tested docking programs were reduced and weakened. The comparisons were focused on the AUC values (see point 3) and mostly to evaluate the specific roles of the applied isomeric and binding spaces. The comparison made in the Conclusions was deleted.

3) Why did the authors used the enrich factor (EF) as the main indicator for the VS evaluation? As I know, the ROC curve and its related metrics (e.g., AUC) are better for this purpose. It is much less dependent on the size difference of screened libraries. As the authors found, four docking software failed to dock some compounds, evaluating VS performance with EF may not suitable for resulting docked libraries with varied numbers of compounds.

The initial analyses of the performances were based on the enrichment factor. This because the quality function, that is optimized by the search algorithm which generates the consensus models, is indeed based on a combination of EF1% values with a skewness descriptor which encodes for the distribution of the active molecules in the entire ranking. That being said and accordingly to the reviewer’s suggestion, a section dealing with the overall comparison of the performed simulations was introduced. This section is primarily based on the analysis of the corresponding AUC values from ROC curves.

4) The authors should also evaluate the accuracy of pose reproducibility. Of course, such evaluation will mainly use SARS-CoV’s 3CL and their inhibitors but may also include some SARS-CoV-2 3CL’s (if any). The authors may collect as many crystal complexes as possible to do this. Although evaluating the VS performance is the key in the manuscript, how useful each software for the pose prediction will be also very important information for further optimization of hit compounds.

The comparison of the generated poses by each docking software goes beyond the objectives of the study, which is focused on the evaluation of the performances when including the proposed space parameters. However, in the section 2.6, a paragraph and a figure were added to analyse and compare the poses computed for the TG-0205221 inhibitor with the recently resolved complex (see above).

Minor comment:

1) Physio-chemical properties should be analyzed by providing some statistical figures like MW, logP, and PSA. Also, some representative chemical structures (2D) should be presented in the main text.

The comparison of some representative physicochemical properties was added under Methods at page 19 and a figure reporting the distribution of some representative physicochemical properties was added in the SI.

2) The authors used too many tables in the manuscript. They could be easily replaced by some well-organized statistical plots. Moreover, please give clear descriptions for each notation in tables or plots. I understand the author may describe them somewhere in the main text, but it is difficult to follow the tables without clear notations.

The total number of the tables was halved in the revised manuscript. Most tables were moved in supporting Information and others replaced by plots (as suggested).

3) How did the authors prepare the protein structures? in “3.1 Library and protein structure preparation”, the author did not give a general procedure. Instead, they described the protein preparation procedures separately for different software. I understand different docking software may have some special requirements, but the protonation states of the key residues in the binding site should be the same for different docking software. The authors should carefully clarify the protein preparation procedure. I suggest giving a general procedure first and then the specialized ones for each software.

The general procedure was briefly discussed in § 3.1 section.

4) In the “method” section, the sequential number of “3.3 PLANTS simulations” is not correct.

Done.

 5) “3.6 Rescoring and consensus analyses” is very lengthy and not well organized. The authors should divide it into several paragraphs and put some flowcharts to facilitate understanding.

The section was subdivided into two parts.

6) How did the authors disperse 478 inhibitors into a database of ~13000? What does “13000 purposely collected safe in man molecules” mean? I think the authors should also provide the statistical plots for physicochemical properties of these decoy molecules as I suggested for the inhibitors. Besides, why did not the authors construct a DUD-like decoy library for the VS evaluation?

The 13000 safe in man molecules were collected by assembling known databases used for repurposing analyses. The inactive molecules were not chosen to be as similar as possible to the active compounds as done in many VS campaigns. We prefer to screen potentially interesting molecules so that the obtained results can be used to assess the performances of the planned VS approaches but also to identify some promising hits (as done in section 3.6). Such a choice poses the problem of possible biasing general differences between active and inactive compounds. However, this risk should be averted by their similar distribution of the physicochemical properties (see above) and, more importantly, by the fact that when used alone, physicochemical and structural descriptors afforded truly unsatisfactory EF1% performances as discussed under Methods.

7) For lines 118 to 119: how do the score averages account for the dynamic profile of the binding process, and the range values encode for the often-neglected entropic component? Could the authors provide references to support their statements? But anyway, I do not think docking scores can give a good estimation for the pose dynamics and binding entropies.

The considerations on the role of the binding space parameters refer to our previous publications (see ref. 22). However,  as suggested by the reviewer, these considerations were weakened in the revised version.

8) For lines 169 to 171: I agree with “the common database” is useful to develop a mixed consensus model. But I do not understand why it is also important for “carefully compare the performances of each docking program”. According to my opinion, failed docked molecules for a specific docking software indicate that the software and/or scoring functions do not “like” those molecules, and those failed molecules can be just considered as the ones with bad docking scores. Those compounds can be still used in the VS evaluation by putting them at the end of the hit list.

The section concerning the comparison of the common database was deleted and only some major results were moved in the supporting information. As said above, the general comparison is based on the entire database using the AUC values. In the revised version, the common database is still used to develop mixed consensus models and to compare the best rankings derived from the performed simulations.

9) For line 194: should the number 479 be corrected to 478?

Done.

10) Please give more clear descriptions for “isomeric space”, “binding space”, “both spaces merged”, “both spaces joint”. Give more details about the ways of their constructions. If necessary, please write down the representative equations.

A detailed explanation for the definition of “isomeric space”, “binding space” and their combination was added at page.

11) Give the details about “n. variables” (what does 1, 2, 3, 4, and 5 mean) together with tables or figures.

The caption of Tables and Figures were enriched to better explain the meaning of the number of variables.

We do believe that the revised version fully answers your comments.

Best regards

Alessandro Pedretti

Reviewer 2 Report

The article by Manfeli et al., presents an interesting concept of ligand binding space to identify the inhibitors of SARS-CoV-2 3CL-Pro. The authors used a combination of four docking protocols and suggested that the docking programs Fred and PLANTS provide the best and the worst predictive power, respectively, while Glide and LiGen show in-between performances. They used a set of compounds that are deemed safe for human use. 

Overall, the authors adopted an extensive computational approach to identify 478 compounds of which 193 has pIC50>60. Overall, the article is worthy of publication in Molecules. However, it needs a massive overhaul. In my opinion, the authors should rephrase the problem from the second paragraph of the "Results' section, which makes a lot more sense than presented in the Abstract and/or Intro sections. The tables are too long and do not provide more information than the text and figures themselves. These Tables are supplementary material and should be moved there.

Additionally, there is a bunch of unnecessary text that makes no sense. For example, the text between lines 68 to 74 does not make any sense. This is just an example. The manuscript is full of this kind of text.

The authors should have the manuscript read by a coronavirus expert. There appear basic problems with the understanding of coronavirus structure. For example, line 56 contains the word 'capsid'. I am not sure where this nomenclature has been adopted from. 

Additionally, ppa1a and ppa1b should be pp1a and pp1ab. 

A serious English edition is needed. 

Author Response

Dear Reviewer,

We thank you for your valuable suggestions. Here is a description of the amendments made in the revised version according to your requests.

The article by Manfeli et al., presents an interesting concept of ligand binding space to identify the inhibitors of SARS-CoV-2 3CL-Pro. The authors used a combination of four docking protocols and suggested that the docking programs Fred and PLANTS provide the best and the worst predictive power, respectively, while Glide and LiGen show in-between performances. They used a set of compounds that are deemed safe for human use. 

Overall, the authors adopted an extensive computational approach to identify 478 compounds of which 193 has pIC50>60. Overall, the article is worthy of publication in Molecules. However, it needs a massive overhaul. In my opinion, the authors should rephrase the problem from the second paragraph of the "Results' section, which makes a lot more sense than presented in the Abstract and/or Intro sections. The tables are too long and do not provide more information than the text and figures themselves. These Tables are supplementary material and should be moved there.

The mentioned first part of the Results was moved in the Introduction. The entire paper was modified as proposed by markedly reducing the number of Tables which are transformed into plots or moved into the Supporting Information.

Additionally, there is a bunch of unnecessary text that makes no sense. For example, the text between lines 68 to 74 does not make any sense. This is just an example. The manuscript is full of this kind of text.

The text was significantly shortened as indicated.

The authors should have the manuscript read by a coronavirus expert. There appear basic problems with the understanding of coronavirus structure. For example, line 56 contains the word 'capsid'. I am not sure where this nomenclature has been adopted from. 

Additionally, ppa1a and ppa1b should be pp1a and pp1ab. 

The part about the coronavirus was amended.

A serious English edition is needed. 

Done.

We do believe that the revised version fully answers your comments.

Best regards

Alessandro Pedretti

Reviewer 3 Report

The manuscript compared different docking engines to screen the SARS-CoV-2 protease inhibitors. I would like to see the inhibition tests of those screened candidates on the top list from the strategies developed by the authors. Otherwise, I do not know if the screening protocols really work or not.

Author Response

Dear Reviewer,

We thank you for your valuable suggestions. Here is a description of the amendments made in the revised version according to your requests.

The manuscript compared different docking engines to screen the SARS-CoV-2 protease inhibitors. I would like to see the inhibition tests of those screened candidates on the top list from the strategies developed by the authors. Otherwise, I do not know if the screening protocols really work or not.

While considering that this is a computational study, the reliability of the performed simulations is indirectly confirmed by the high number of inhibitors of SARS-CoV protease included in the top lists. Moreover, the recent literature was analysed to see if other top ranked molecules were reported as SARS-CoV-2 3CL-Pro inhibitors. Such a search evidenced  a significant number of heterogeneous molecules, the SARS-CoV-2 3CL-Pro inhibition activity was recently reported. All these compounds were discussed in the Results.            

We do believe that the revised version fully answers your comments.

Best regards

Alessandro Pedretti

Round 2

Reviewer 1 Report

The improved version is eligible for publishing.

Reviewer 2 Report

The authors have addressed the comments. Therefore, I am happy to endorse the paper for the publication.

Reviewer 3 Report

I appreciate the revision from authors to improve the quality and impact of the manuscript. I agree to accept the manuscript for publication.